# Effect of goal-oriented prenatal education on birth preparedness, complication readiness and institutional delivery among semi-urban pregnant women in Nigeria: A quasi-experimental study

**Margaret Omowaleola Akinwaare****\*, Oyeninhun Abimbola Oluwatosin**

Faculty of Clinical Sciences, Department of Nursing, College of Medicine, University of Ibadan, Ibadan, Nigeria

\* margaretakinwaare@gmail.com

## Abstract

### Background

High maternal mortality has been associated with inadequate Birth preparedness and Complication Readiness (BPCR) and non-institutional delivery in developing countries. Therefore, there is a need for proven interventions that will improve BPCR and institutional delivery to reduce maternal mortality. Therefore, this study evaluated the effects of Goal-Oriented Prenatal Education (GOPE) on pregnant women's BPCR and institutional delivery.

### Methods

The study adopted a quasi-experimental two-group pre and post-test design. Two Local Government Areas (LGAs) were randomly selected from the six semi-urban LGAs in Ibadan. These LGAs were randomized into an intervention and control group. Two Primary Healthcare Centres (PHCs) were randomly selected from each LGA, and 400 pregnant women who registered for antenatal care in the selected PHCs, and met the inclusion criteria were purposively selected to participate in the study. A validated questionnaire and checklist were adapted for data collection at baseline and post-intervention. The pregnant women in the intervention group received GOPE focusing on knowledge and attitude to BPCR. Participants' place of birth was documented at delivery. Data were analyzed using descriptive statistics, and the Mann-Whitney U test at α0.05.

### Results

Good knowledge of BPCR was found in 65.5% of pregnant women at baseline and 91.8% post-intervention. Good BPCR practice was found in 95.3% and 73.1% of women in the intervention and control groups respectively. At delivery, 93.5% and 53.5% had institutional delivery in the intervention and control groups respectively. A significant difference (p<0.001) was observed in BPCR knowledge and attitude post-intervention, as well as in

**Data Availability Statement:** All relevant data are within the paper.

**Funding:** MA received the award "This research is supported by Consortium for Advanced Research Training in Africa (CARTA). CARTA is jointly led by the African Population and Health Research Centre and the University of the Witwatersrand (https://cartafrica.org/) and funded by the Carnegie Corporation of New York (Grant No. G-19-57145) (https://www.carnegie.org/), Sida (Grant No. 54100113) (https://www.sida.se/en), Uppsala Monitoring Centre, Norwegian Agency for Development Cooperation (Norad) (https://who-umc.org/), and by the Wellcome Trust [reference no. 107768/Z/15/Z] (https://who-umc.org/) and the UK Foreign, Commonwealth & Development Office, with support from the Developing Excellence in Leadership, Training and Science in Africa (DELTAS Africa) programme. The statements made and views expressed are solely the responsibility of the Fellow. For the purpose of open access, the author has applied a CC BY public copyright licence to any Author Accepted Manuscript version arising from this submission." The funders had no role in study design, data collection and analysis, decision to publish, or preparation of the manuscript.

**Competing interests:** The authors have declared that no competing interests exist

BPCR practice and institutional delivery between women in the intervention and control group.

## Conclusions

Goal-oriented prenatal education improved birth preparedness and complication readiness as well as institutional delivery among pregnant women. This should be integrated into routine prenatal education in Nigeria.

## Introduction

Maternal mortality is still a global public health problem, with a global ratio of 211 maternal deaths per 100,000 live births [1]. The lifetime risk of death from pregnancy and delivery-related complications has been reported to be as high as 462 per 100,000 live births in low and middle-income countries as against 11 per 100,000 live births in high-income countries [1]. This is supported by Girum et al [2] who reported maternal death as a major problem in sub-Saharan Africa where there is a very slow decline in the maternal mortality ratio. In Nigeria, the maternal mortality ratio is 512 per 100,000 live births, meaning that five women out of 1000 are likely to die in relation to pregnancy, childbirth, and post-childbirth [3]. Therefore, pregnancy and childbirth periods are critical times in the life of a woman, hence, the need for special care to ensure positive pregnancy outcomes [4].

The risk of maternal death and morbidity is much higher in Nigeria in comparison to other countries in sub-Saharan Africa [1]. Low accessibility to skilled care during pregnancy and delivery resulting from inadequate birth preparedness and complication readiness (BPCR) has been associated with maternal deaths [1]. Thus, birth preparedness and complication readiness (BPCR) is a key strategy to receiving skilled care during pregnancy and delivery [5], especially in developing countries like Nigeria. It is therefore imperative to provide interventions that will improve BPCR among pregnant women.

Moinuddin et al [6] identified BPCR as an essential and useful strategy with many advantages such as motivating women to plan for delivery and to receive care from a skilled provider at delivery. A major approach to reducing maternal mortality is to make plans to accommodate both normal and complicated processes from pregnancy to childbirth. Onoh et al [7] reported that BPCR entails both the knowledge and actions that enhance maternal health. This is corroborated by Imaralu et al [8], who documented that BPCR involves planning for natural birth and at the same time planning for actions necessary in case there is an emergency. However, interventions to improve BPCR have not been well documented.

The BPCR was also described as planning and preparation in advance for delivery to improve maternal health outcomes [8]. It can be approached at different levels which include; the level of the individual, the family level, the community level, the health facility level, the service provider level, and the policymaker's level [9]. The concept of BPCR is considered to consist of three main domains which are the knowledge of obstetric danger signs, knowledge of skilled birth attendants as well as identifying one for delivery, and knowledge of elements of BPCR [10]. Therefore, it is important to look at BPCR from its three domains and among the identified stakeholders.

A report from a previous study among pregnant women attending antenatal clinics in Nigeria [11] showed a significant association between unplanned home delivery and inadequate BPCR. Therefore, this raised questions about the content of prenatal education, as well as the

modalities of providing prenatal education. Hence, inappropriate prenatal education could lead to inadequate BPCR which consequently will result in unskilled birth attendance, worsening the already high maternal death in Nigeria and other low and middle-income countries (LMICs). Thus, the content and modalities of delivery of prenatal education during prenatal care are of great concern in ensuring adequate BPCR among pregnant women. Lack of adequate information on BPCR during prenatal education is associated with maternal death [12]. Thus, there is an association between prenatal education and the health behavior of pregnant women, but there is a dearth of information on the inclusion of BPCR education in prenatal education, and there is limited information on the association between prenatal education and adequate BPCR with its resultant institutional delivery among pregnant women in Nigeria.

Therefore, this study evaluated the effect of Goal-Oriented Prenatal Education (GOPE) on BPCR and institutional delivery among pregnant women in selected local government areas of Ibadan, Nigeria.

## Methods

### Study aim

The overall aim of this study is to evaluate the effects of goal-oriented prenatal education on birth preparedness and complication readiness of pregnant women and institutional delivery in selected primary healthcare facilities in semi-urban areas of Ibadan, Nigeria.

### Study design

A quasi-experimental two-group pre-post design was adopted for the study. Details of the study flow chart are shown in Fig 1.

### Study setting

This study was conducted in Oluyole and Akinyele local government areas (LGAs) of Ibadan, Nigeria. These are two out of the six semi-urban LGAs in Ibadan, Nigeria. Two Primary Healthcare Centers (PHCs) with the highest antenatal client flow were selected from each of the selected LGAs. The PHCs run antenatal clinics on a weekly basis, with an average of 80 clients.

### Sampling technique

A three-stage sampling technique was used for the study. The first stage involved simple random selection (through balloting) of two out of six semi-urban LGAs in Ibadan. Two selected LGAs were Oluyole and Akinyele LGAs. These two selected LGAs were randomized to intervention and control groups also through balloting. The second stage included drawing a list of all the primary healthcare centers (PHCs) in each local government, Ojoo and Moniya (out of 13) PHCs were purposively selected based on the high client flow from Akinyele LGA while Adaramagbo and Odo-ona elewe (out of 28) PHCs were randomly selected out of four PHCs with highest client flow from Oluyole LGA. At the third stage, all antenatal clinic attendees who met the inclusion criteria at each visit were purposively selected to participate in the study.

### Sample size determination

The sample size was calculated according to the sample for proportions formula by Charan et al [13] with a 95% confidence level and 0.05 precision to yield a representative sample

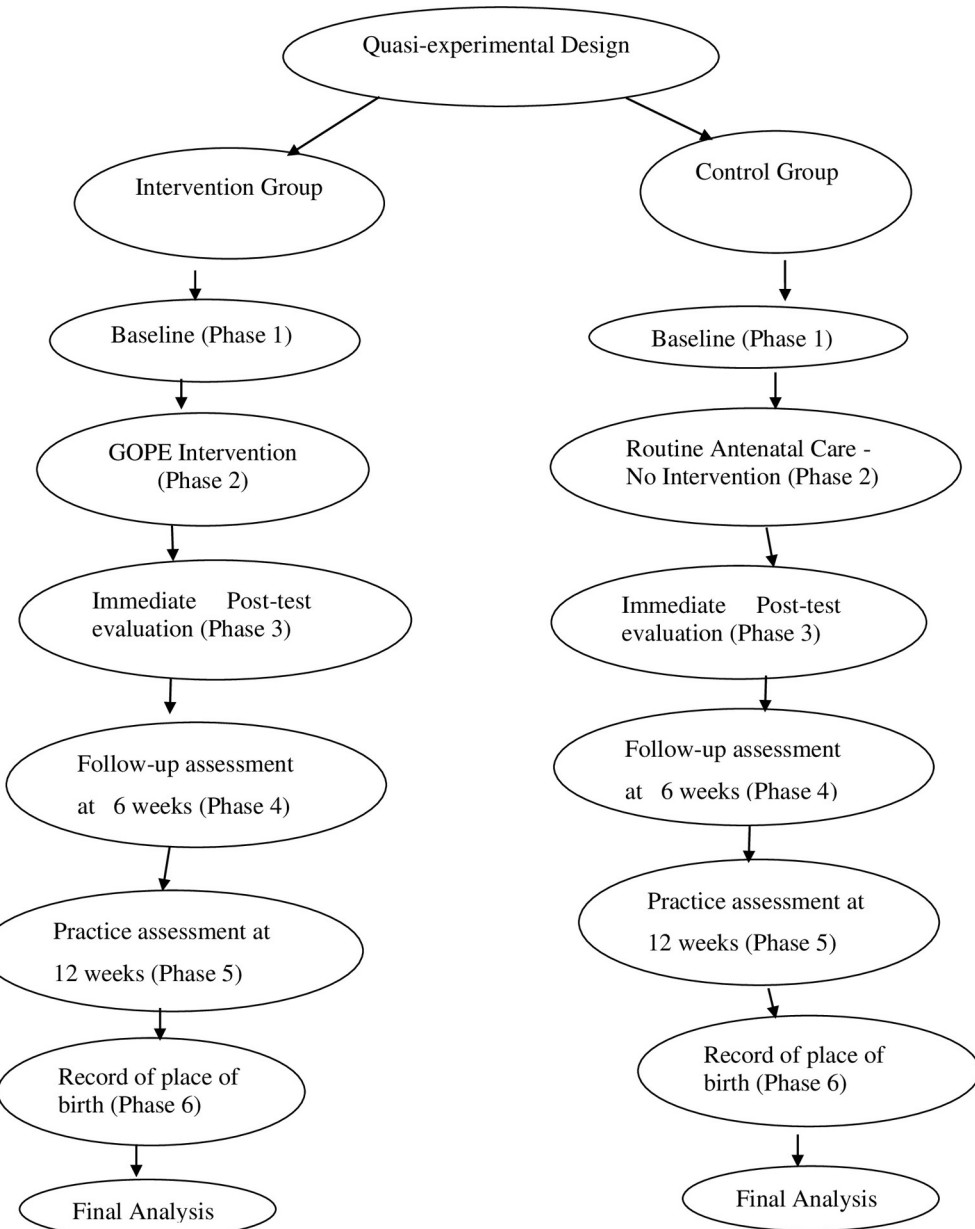

**Fig 1. Study design flow chart showing what was done during the six phases of the study.**

using the formula below:

$$N = 2(Z\alpha + Z\beta)^2 P(1 - P)/(P1 - P2)^2$$

Where:

N–sample size

Zα –standard normal deviate corresponding to the null hypothesis = 1.96 (from Z table) at type 1 error of 5%

Zβ –standard normal deviate corresponding to the alternative hypothesis = 0.842 (from Z table) at 80% power

P1 –the proportion of practice of birth preparedness and complication readiness in the previous study = 35% [14]

P2—the proportion of practice of birth preparedness and complication readiness desired in current study = 50%

P = pooled proportion = [proportion in previous study (P1) + proportion in current study (P2)] / 2

N = $\underline{2(1.96+0.842)2\ 42.5(1–42.5)\ /}$ $(35–50)^2 = 170$

Based on the calculated sample size, a minimum of 170 pregnant women were to be studied. Assuming an additional loss to follow-up and a non-response rate of 10%, the minimum number of pregnant women to be studied was calculated to be 187, which was rounded up to 200 in each control and intervention group.

## Characteristics of participants

Four hundred pregnant women were recruited across the selected health facilities to participate in the study following appropriate inclusion criteria which included; pregnant women who have signed up for antenatal care at one of the participating health facilities, and pregnant women with an estimated gestational age of 20 to 24 weeks.

## Instrument development

The instrument for the study included a questionnaire, a checklist, and intervention tools (BPCR flip chart and BPCR card). The study adopted a validated, semi-structured/structured questionnaire from 'Monitoring BPCR tools and indicators for maternal and newborn health' developed by Johns Hopkins Program for International Education in Gynaecology and Obstetrics [15]. The instrument was modified following a rigorous review of literature based on relevance to the cultural values of the study setting. An observational checklist was developed by the researcher after a thorough literature review. It consists of 11 items used to assess expected actions indicating BPCR practice. The reliability of the instrument was conducted to assess the psychometric properties and ensure the adaptability of the instrument to the study setting. Also, back-to-back translation of the instrument (English and Yoruba language) was done to ascertain its reliability of the instrument. The correlation coefficient of scores was obtained to evaluate the test for stability over time. The result of the reliability test was found to be 0.8. Hence, the instrument was considered to be reliable.

The intervention tools were developed by the researcher. These are BPCR cards and BPCR flip charts, both were developed in English and Yoruba (local dialect) language.

**BPCR card.** The BPCR card was designed to complement the usual antenatal card, a 6-page, and 3-fold card. The first page provided information on the research study objective and briefly introduced the researcher, including the researcher's contact. The second page is for documentation of the personal and obstetric data of the participants. Such as name, expected date of delivery, and appointment date. The third, fourth, and fifth pages are displayed in pictures and words, likely health problems during pregnancy that require a prompt report to the health facility. The last (sixth) page has a list of elements of BPCR. The BPCR card is handy to remind them of the information given during educational intervention.

**BPCR flip chart.** The BPCR flip chart is an 8-page chart. It was compiled and designed by the researcher. The 8-page chart contained information that was presented in simple pictures to capture the attention of the respondents and to facilitate learning among the educated, less educated, and uneducated participants. The content focuses on obstetric danger signs, elements of BPCR, and characteristics of skilled birth attendants. The participants were instructed

to hang the chart in their various homes so that it can also provide information for their spouses at home and can also remind them of what they have been taught.

## Data collection procedure

The study was done, starting from the pre-intervention, the intervention, and the post-intervention. The study was conducted in six phases (the intervention phase inclusive) while data collection and analysis were done in five phases (Baseline–P0, Immediate post-intervention–P1, six weeks post-intervention–P2, twelve weeks post-intervention–P3, and at delivery–P4).

At the pre-intervention stage/planning stage, the research assistants were trained, a pilot study was conducted and baseline data were collected.

**Phase 1: Participants' recruitment and Baseline survey (P0).** The recruitment of the study participants spanned from March 2019 to January 2020. All pregnant women who met the inclusion criteria were recruited during the antenatal clinic, and scheduled for intervention at a pre-determined day and venue (intervention group), while the control group had routine prenatal education. Both groups were followed up for continued data collection till their delivery. This process of recruitment and follow-up continued until the researcher got the desired sample size of 200 participants in each of the intervention group sites and control group sites.

At recruitment, each participant was given an identification number, and the validated questionnaire was administered by the trained research assistants. Also, the phone numbers of all participants were documented to remind them of the intervention date and/or follow-up interview date.

**Phase 2: Intervention.** *The Intervention Process.* The intervention group was exposed to goal-oriented prenatal education (GOPE), focusing on BPCR with the use of a BPCR flip chart. The content was taught by the researcher for 20 minutes and there was another interactive session of 10 minutes. The GOPE was integrated into routine prenatal education. Participants were informed of the next post-intervention data collection.

**Phase 3: Immediate post-intervention (P2).** Immediate post-intervention data collection was done two weeks after the intervention in the two groups using the questionnaire. Each participant reported back for data collection as scheduled. The date for the data collection was fixed to coincide with their antenatal clinic day.

**Phase 4: Six weeks post-intervention follow-up (P3).** This included data collection with the use of a questionnaire six weeks after the intervention. This was meant for a follow-up assessment to assess retained memory over some time after the intervention. Each participant reported back for data collection at the clinic site as scheduled. The date for the data collection was planned to coincide with their antenatal clinic day.

**Phase 5: Twelve weeks follow up.** This included data collection with the use of a checklist at twelve weeks after the intervention. This was meant to assess the necessary actions taken or not taken by pregnant women toward safe delivery at a time very close (four to six weeks) to their expected date of delivery. The participants reported back to the clinic for data collection as scheduled. The date for the data collection was planned to coincide with their antenatal clinic day.

**Phase 6: Delivery monitoring.** A structured weekly telephone call was put across to all participants from 36 weeks of estimated gestational age till delivery. The main purpose of the telephone call was to ask whether or not the participant had delivered, and where the participant delivered. Many of them also called the researcher to inform the researcher of their delivery and the site of delivery. This was to document their site of birth which is a determinant of skilled birth attendance.

For ethical reasons, debriefing at the control sites was done after the completion of data collection for the study.

## Data analysis

Preliminary checking of the questionnaires was carried out for errors. The data were entered into IBM—SPSS version 22. Percentages were used to summarize categorical variables, while mean, median, and standard deviation were used to summarize continuous variables. The 50th percentile of the knowledge score was used in both the intervention and control groups for the categorization of variables. Bivariate analysis was done using a chi-square test between the control and intervention groups at baseline and post-intervention. The significance level was set at $p < 0.05$. Relevant tables and figures were used in presenting some of the results.

## Ethical considerations

Ethical approval number UI/EC/18/0629 was obtained from the University of Ibadan/University College Hospital institutional review committee before the commencement of data collection. The objectives of the study were explained to the study participants, and verbal informed consent was obtained. They were assured of the confidentiality of the information received and the right to opt out at any point of the study without any consequence.

# Results

## Recruitment and follow-up study flow

Four hundred pregnant women were recruited for the study. However, not all were available for data collection at various phases of the study as shown in Fig 2.

## Sociodemographic characteristics of study participants

The socio-demographic characteristics of the women who participated in this study are as presented in Table 1. The mean age(±SD) is 27.4(±4.9) and 27.1(±5.1) for women in the intervention and control group respectively. The study reveals the obstetric history of the participants, disaggregated by the study groups (Table 1).

## Participants' knowledge of obstetric danger signs

The level of good knowledge of obstetric danger signs is similar in both the intervention group (57%) and control group (58%) at baseline (P0). However, post-intervention, there was a statistically significant association between the level of knowledge of obstetric danger signs and the study group (p,0.001). Details are presented in Table 2.

## Participants' knowledge of elements of BPCR

No statistical association was observed between the knowledge level of elements of BPCR and the study group the women belonged to (p = 0.178) at baseline (P0). However, post-intervention, there was a statistical association between the knowledge level of the elements of BPCR and the study group (p<0.001). Details are shown in Table 3.

## Participants' knowledge/recognition of skilled birth attendant

The level of knowledge/recognition of skilled birth attendant was found to be similar at baseline among the women in the intervention (87%) and control (89.5%) groups (Table 4).

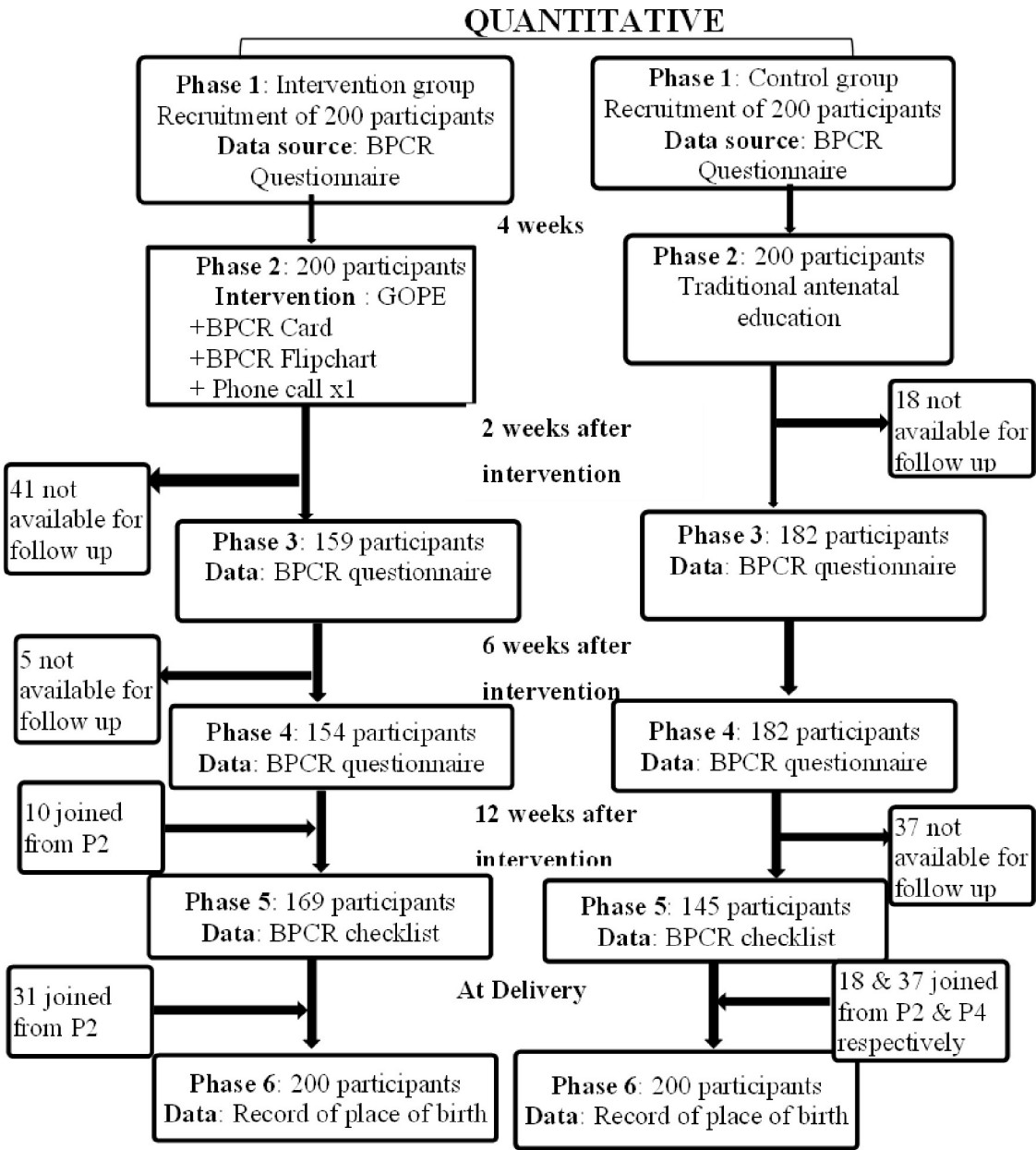

**Fig 2. Study follow-up flow chart showing participants and data collection at each phase of the study.**

However, the knowledge/recognition was found to be better at P1 (Table 4) among women in the intervention group (97%) than women in the control group (91%).

## Participants' overall knowledge of BPCR

The knowledge of the three domains of BPCR knowledge (knowledge of obstetric danger signs, knowledge of elements of BPCR, and knowledge/recognition of SBA) was pooled together to assess the overall knowledge of BPCR, the mean score was found to be 45.58 (SD ±11.71) and 36.03 (SD±14.52) among women in intervention and control group respectively at P2 as presented on Table 5.

**Table 1. Socio-demographic characteristics of study participants.**

| Variable | Intervention N (%) | Control N (%) | Total N (%) |
|---|---|---|---|
| **Age** | | | |
| Less than 20 years old | 8 (4.0) | 10 (5.0) | 18 (4.5) |
| 20–29 years old | 125 (62.5) | 137 (68.5) | 262 (65.5) |
| 30–39 years old | 65 (32.5) | 50 (25.0) | 115 (28.7) |
| 40–49 years old | 2 (1.0) | 3 (1.5) | 5 (1.3) |
| **Marital Status** | | | |
| Single | 14 (7.0) | 13 (6.5) | 27 (6.8) |
| Married | 186 (93.0) | 187 (93.5) | 373 (93.2) |
| **Highest level of education** | | | |
| No formal education | 6 (3.0) | 3 (1.5) | 9 (2.3) |
| Primary education | 15 (7.5) | 14 (7.0) | 29 (7.3) |
| Secondary education | 109 (54.5) | 134 (67.0) | 243 (60.8) |
| Tertiary education | 70 (35.0) | 49 (24.5) | 119 (29.8) |
| **Religion** | | | |
| Christianity | 104 (52.0) | 67 (33.5) | 171 (42.8) |
| Islam | 96 (48.0) | 133 (66.5) | 229 (57.2) |
| **Ethnicity** | | | |
| Yoruba | 178 (89.0) | 190 (95.0) | 368 (92.0) |
| Igbo | 10 (5.0) | 7 (3.5) | 17 (4.3) |
| Hausa | 6 (3.0) | 1 (0.5) | 7 (1.8) |
| Others | 6 (3.0) | 2 (1.0) | 8 (2.0) |
| **Monthly Income** | | | |
| Below 20,000 naira | 135 (67.5) | 137 (68.5) | 272 (68.0) |
| 20,000–40,00 naira | 44 (22.0) | 51 (25.5) | 95 (23.8) |
| 40,000–60,000 naira | 17 (8.5) | 9 (4.5) | 26 (6.5) |
| 60,000–80,000 naira | 2 (1.0) | 2 (1.0) | 4 (1.0) |
| Above 80,000 naira | 2 (1.0) | 1 (0.5) | 3 (0.8) |
| **Participant's occupation** | | | |
| Clerical/Skilled Artisans | 77 (38.5) | 75 (37.5) | 152 (38.0) |
| Sales and services | 68 (34.0) | 70 (35.0) | 138 (34.5) |
| Professional/Managerial | 29 (14.5) | 30 (15.0) | 59 (14.8) |
| Unemployed | 26 (13.0) | 25 (12.5) | 51 (12.8) |
| **First pregnancy** | | | |
| Yes | 94 (47.0) | 94 (47.0) | 188 (47.0) |
| No | 106 (53.0) | 106 (53.0) | 212 (53.0) |
| **Stage of pregnancy at ANC registration** | | | |
| First trimester | 39 (19.5) | 55 (27.5) | 94 (23.5) |
| Second trimester | 157 (78.5) | 142 (71.0) | 299 (74.8) |
| Third trimester | 4 (2.0) | 3 (1.5) | 7 (1.8) |
| **Number of ANC attendance before recruitment** | | | |
| None | 33 (16.5) | 7 (3.5) | 40 (10.0) |
| 1–3 times | 152 (76.0) | 151 (75.5) | 303 (75.8) |
| 4 or more times | 15 (7.5) | 42 (21.0) | 57 (14.2) |
| **Number of deliveries** | | | |
| None | 94 (47.0) | 94 (47.0) | 188 (47.0) |
| One | 47 (23.5) | 45 (22.5) | 92 (23.0) |

*(Continued)*

**Table 1.** (Continued)

| Variable | Intervention N (%) | Control N (%) | Total N (%) |
|---|---|---|---|
| Two | 30 (15.0) | 33 (16.5) | 63 (15.8) |
| Three | 20 (10.0) | 18 (9.0) | 38 (9.5) |
| Four | 5 (2.5) | 9 (4.5) | 14 (3.5) |
| More than 4 | 4 (2.0) | 1 (0.5) | 5 (1.3) |

Notably, 66.5% of women in the intervention and 60.5% in the control group had good knowledge of BPCR at baseline (P0). However, at immediate post-intervention (P1), 146 (91.8%) of women in the intervention group had good knowledge of BPCR while 135 (74.2%) had good knowledge among women in the control group (Table 6).

### Participants' practice of BPCR

The result from the study showed that a statistically significant association exists between the level of BPCR practice and the study group (p < 0.001 as shown in Table 7.

### Participants' site of birth

The result of the place of birth as presented in Fig 3 shows that 187(93.5%) of women in the intervention group had institutional births while 107(53.5%) of the control group had institutional births. A very high number 76(38.0%) of the control group gave birth at home while only 6(3.0%) of the intervention group gave birth at home.

## Discussion

This study has proven the effectiveness of goal-oriented prenatal education on all the three components of BPCR, as well as on institutional delivery among pregnant women. The result findings revealed better outcomes in terms of the knowledge and practice of BPCR, and a higher number of institutional delivery among the pregnant women in the intervention group compared with their counterpart in the control group.

The pregnant women who participated in this study had an average level of knowledge of obstetric danger signs at baseline. This is consistent with the findings of other descriptive studies conducted in Ethiopia [16] and in Nigeria [17]. However, after the intervention, the immediate assessment shows a better improvement among the women in the intervention group. The assessment at six weeks post-intervention showed that the pregnant women in the intervention group had a better knowledge of obstetric danger signs then their counterpart in the control group. This improvement in the level of knowledge post-intervention is similar to a

**Table 2.  Participants' level of knowledge of obstetric danger signs pre and post-intervention.**

| Level of knowledge | P0 | | | | | P1 | | | | | P2 | | | | |
|---|---|---|---|---|---|---|---|---|---|---|---|---|---|---|---|
| | Intv. n = 200 | Ctrl n = 200 | Total | $X^2$ | P-value | Intv. n = 159 | Ctrl n = 182 | Total | $X^2$ | P-value | Intv. n = 154 | Ctrl n = 182 | Total | $X^2$ | P-value |
| Good knowledge | 114 (57.0) | 116 (58.0) | 230 (57.5) | 0.04 | 0.840 | 143 (89.3) | 128 (70.2) | 270 (79.2) | 18.54 | < 0.001 | 131 (85.1) | 116 (63.7) | 247 (73.5) | 19.49 | < 0.001 |
| Poor knowledge | 86 (43.0) | 84 (42.0) | 170 (42.5) | | | 17 (10.7) | 54 (29.8) | 71 (20.8) | | | 23 (14.9) | 66 (36.3) | 89 (26.5) | | |

**Table 3. Participants' level of knowledge of elements of BPCR pre and post-intervention.**

| Level of knowledge | P0 | | | | | P1 | | | | | P2 | | | | |
|---|---|---|---|---|---|---|---|---|---|---|---|---|---|---|---|
| | Intv. n = 200 | Ctrl n = 200 | Total | X$^2$ | P-value | Intv. n = 159 | Ctrl n = 182 | Total | X$^2$ | P- value | Intv. n = 154 | Ctrl n = 182 | Total | X$^2$ | P- value |
| Good knowledge | 172 (86.0) | 162 (81.0) | 334 (83.5) | 1.82 | 0.178 | 156 (98.1) | 148 (81.3) | 304 (89.1) | 24.75 | < 0.001 | 152 (98.7) | 148 (81.3) | 300 (89.3) | 26.35 | < 0.001 |
| Poor knowledge | 28 (14.0) | 38 (19.0) | 66 (16.5) | | | 3 (8.1) | 34 (18.7) | 37 (10.9) | | | 2 (1.3) | 34 (18.7) | 36 (10.7) | | |

quasi-experimental study [18] conducted in Tanzania which reported an increase of one-third and one-twentieth in the level of knowledge of obstetric danger signs in the intervention and control group respectively. The differences between the intervention and the control group could be due to the effect of goal-oriented prenatal education (GOPE). Masoi et al [18] also supported this assumption in their study.

Also, the pregnant women reported awareness of BPCR at recruitment to the study. Therefore, at recruitment, the majority of them had good knowledge of BPCR. This is similar to other studies conducted in India [19], and Ethiopia [20]. On the contrary, other descriptive studies in Nigeria [21,22] reported poor knowledge of elements of BPCR among pregnant women attending antenatal clinics. This variation could be due to the information and content of prenatal education received during the antenatal clinic. However, in post-intervention, a greater proportion of the pregnant women in the intervention group had a better knowledge of elements of BPCR than the pregnant women in the control group. The differences between the intervention and the control group could be attributed to the effect of GOPE received by the intervention group which proves effective in enhancing pregnant women's knowledge of elements of BPCR. Such intervention has been reported to be effective among rural dwellers as reported by Timsa et al [23], and Adam et al [24] in different studies conducted in Uganda and Kenya respectively. The increase in the knowledge of pregnant women in the control group might also be attributed to administering the same questionnaire repeatedly which might have raised their curiosity to find out the correct answers to those questions. Overall, women in the intervention group had more knowledge of elements of BPCR than those in the control group. It could therefore be implied that exposing pregnant women to effective interventions such as GOPE will help to improve their knowledge on issues relating to maternal health and positive pregnancy outcome.

Furthermore, findings from this study revealed that the majority of the pregnant women who participated in the study, both the intervention and control groups had good knowledge and could recognize an SBA at recruitment and at six weeks post-intervention. Almost all of

**Table 4. Participants' level of knowledge and recognition of skilled birth attendant's pre and post-intervention.**

| SBA recognition | P0 | | | | | P1 | | | | | P2 | | | | |
|---|---|---|---|---|---|---|---|---|---|---|---|---|---|---|---|
| | Intv. n = 200 | Ctrl n = 200 | Total | X$^2$ | P-value | Intv. n = 159 | Ctrl n = 182 | Total | X$^2$ | P- value | Intv. n = 154 | Ctrl n = 182 | Total | X$^2$ | P- value |
| Good recognition | 174 (87.0) | 179 (89.5) | 353 (88.3) | 0.60 | 0.438 | 154 (96.9) | 165 (90.7) | 319 (93.5) | 5.40 | 0.020 | 145 (94.2) | 166 (91.2) | 311 (92.6) | 1.05 | 0.305 |
| Poor recognition | 26 (13.0) | 21 (10.5) | 47 (11.8) | | | 5 (3.1) | 17 (9.3) | 22 (6.5) | | | 9 (5.8) | 16 (8.8) | 25 (7.4) | | |

**Table 5. Participants' knowledge mean score of BPCR pre and post-intervention.**

| | Intervention Mean ± SD | Control Mean ± SD | P-value (t-test) |
|---|---|---|---|
| **Knowledge of obstetric danger signs** | | | |
| Baseline (P0) | 18.3 ± 12.6 | 18.12 ± 13.2 | 0.898 |
| Immediate post intervention (P1) | 28.6 ± 9.2 | 19.6 ± 12.4 | < 0.001 |
| Six weeks post intervention (P2) | 27.1 ± 10.8 | 19.6 ± 13.3 | < 0.001 |
| **Knowledge on elements** | | | |
| Baseline (P0) | 7.1 ± 1.6 | 6.9 ± 2.8 | 0.253 |
| Immediate post intervention (P1) | 8.2 ± 0.9 | 6.9 ± 1.7 | < 0.001 |
| Six weeks post intervention (P2) | 8.1 ± 0.9 | 6.9 ± 1.7 | < 0.001 |
| **Birth attendant recognition** | | | |
| Baseline (P0) | 9.2 ± 2.2 | 9.2 ± 1.9 | 0.941 |
| Immediate post intervention (P1) | 10.7 ± 1.6 | 9.5 ± 1.9 | < 0.001 |
| Six weeks post intervention (P2) | 10.3 ± 1.8 | 9.5 ± 1.8 | < 0.001 |
| **BPCR knowledge** | | | |
| Baseline (P0) | 34.6 ± 13.9 | 34.2 ± 14.3 | 0.794 |
| Immediate post intervention (P1) | 47.4 ± 5.6 | 37.9 ± 13.8 | < 0.001 |
| Six weeks post intervention (P2) | 45.6 ± 11.7 | 36.0 ± 14.5 | < 0.001 |

them had good knowledge of SBA and could recognize them. This could be related to their good knowledge of elements of BPCR as supported by a study conducted in Uganda [25]. Florence et al [25] reported that the majority of the pregnant women who participated in the study had good knowledge of BPCR as they were able to identify SBA. Thus, this study implied that good knowledge/ recognition of SBA is significantly associated with good knowledge of BPCR. However, women in the intervention group had a better knowledge/recognition of an SBA than those in the control group. Therefore, exposure of pregnant women in the intervention group to GOPE improved their knowledge/recognition of SBA.

This study pooled the three domains of BPCR (knowledge of obstetric danger signs, knowledge of elements of BPCR, and knowledge/recognition of skilled birth attendants) together in reporting BPCR in totality. Therefore, the overall level of good BPCR was similar among study participants in both the intervention group and the control group at baseline. However, in post-intervention, more participants in the intervention group have good knowledge of BPCR than their counterparts in the control group. This clear difference was attributed to the exposure of the intervention group to GOPE. The effectiveness of various interventions such as GOPE has been predicted, however, this has been confirmed and has become evidence-based with the findings of this study.

**Table 6. Participants' level of knowledge of BPCR pre and post-intervention.**

| | P0 | | | | | P1 | | | | | P2 | | | | |
|---|---|---|---|---|---|---|---|---|---|---|---|---|---|---|---|
| Level of knowledge | Intv. n = 200 | Ctrl n = 200 | Total | $X^2$ | P-value | Intv. n = 159 | Ctrl n = 182 | Total | $X^2$ | P-value | Intv. n = 154 | Ctrl n = 182 | Total | $X^2$ | P-value |
| Good knowledge | 133 (66.5) | 121 (60.5) | 254 (63.5) | 1.55 | 0.213 | 146 (91.8) | 135 (74.2) | 281 (82.4) | 18.23 | < 0.001 | 134 (87.0) | 122 (67.0) | 256 (76.2) | 18.36 | < 0.001 |
| Poor knowledge | 67 (33.5) | 79 (39.5) | 146 (36.5) | | | 13 (8.2) | 47 (25.8) | 60 (17.6) | | | 20 (13.0) | 60 (33.0) | 80 (23.8) | | |

**Table 7. Practice of BPCR between the study groups.**

| Variable | Intervention (n = 169) | Control (n = 145) | Total | $X^2$ | P-value |
|---|---|---|---|---|---|
| Good BPCR practice | 161 (95.3) | 106 (73.1) | 267 (85.0) | 30.12 | < 0.001 |
| Poor BPCR practice | 8 (4.7) | 39 (26.9) | 47 (15.0) | | |

Furthermore, this study supports the presumption that adequate preparation for birth improves with educating pregnant women on BPCR during prenatal classes [26]. Thus, a greater percentage of pregnant women in the intervention group compared to those in the control group have done what was expected of them before delivery to demonstrate their preparedness for birth. Their actions included; identification of SBA with whom they intended to deliver, choosing a birth location, arranging for transportation to a health facility whenever labour commences, buying necessary birth supplies as required by the health facility, arrangement for a temporary family carer while away for delivery and saving money for skilled care at childbirth. Women in the intervention group had a better BPCR practices than those in the control group. Similarly, using the BPCR practice scores; a statistical difference was observed in the actual practice score between the two groups. The average score among women in the intervention group was higher than their counterparts in the control group. Therefore, it can be concluded that exposing pregnant women to effective interventions like GOPE improved the BPCR practice among pregnant women.

In addition, institutional delivery, which is referred to as birth at either private or public health facilities has been reported to have very high chances of being assisted by an SBA,

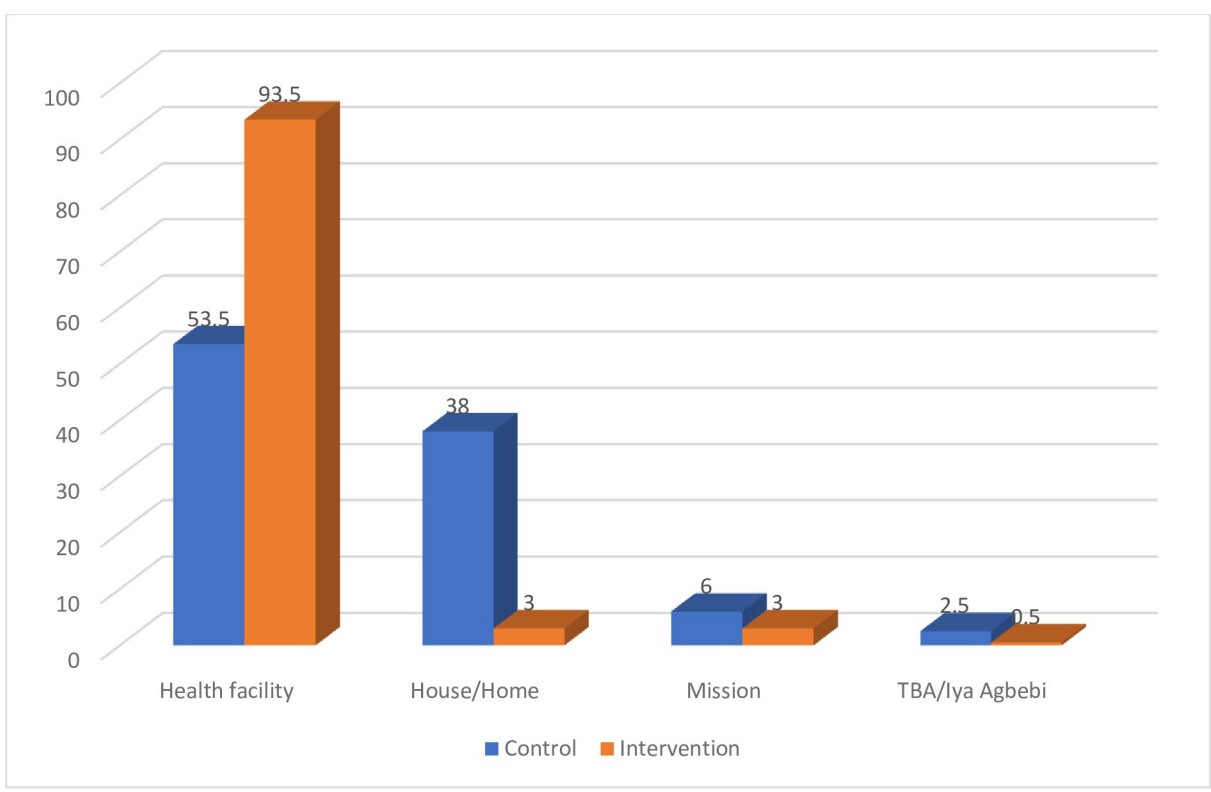

**Fig 3. Participants' site of birth.**

especially in developing countries like Nigeria. Thus, because of the risk associated with pregnancy, institutional or hospital delivery has been recommended for all pregnant women in developing countries like Nigeria [11]. In this study, it could also be implied that a good number of the women in the control group had institutional delivery when compared with the low level of institutional delivery in Nigeria according to NDHS [3]. However, NDHS reports by the states in Nigeria indicated that institutional delivery is higher in southwestern Nigeria when compared to the northern states. This could probably be responsible for the relatively high institutional delivery among women in the control group. However, almost all pregnant women in the intervention group had their deliveries at the health facility. This contradicts the findings of many descriptive studies conducted in Africa. For example, Kidanu et al, [27] reported that one-third of institutional delivery among recently delivered women in Dembecha district, Northwest Ethiopia, Bishanga et al [28] reported half of the participants had institutional delivery among recently delivered mothers in Mara, Tanzania while the Nigeria Demographic and Health Survey [3] reported institutional delivery of about one-third among Nigerian women.

Therefore, the high percentage of pregnant women who have institutional delivery in the intervention group could be attributed to the exposure of the women to GOPE. This study shows further that those women in the intervention group were much more likely to have their deliveries in a health facility, as compared to those in the control group. This is consistent with a study conducted in a rural community of Ethiopia by Eshete et al, [29], who reported that maternal knowledge of the benefit of institutional delivery as contained in GOPE and BPCR education was significantly associated with institutional delivery. Hence, it could be deduced that GOPE is an effective intervention in form of provision of BPCR education for pregnant women during the antenatal clinic and is significantly associated with institutional delivery.

## Strengths of the study

This study was able to provide a piece of evidence-based information that goal-oriented prenatal education is a key strategy to improve BPCR and institutional delivery among pregnant women. The study was able to retain the majority of the pregnant women who participated in the study, and the loss to follow-up was very minimal. Also, the study showed that the pregnant women in both the intervention and control groups share similar socio-demographic characteristics and level of knowledge pre-intervention. This gave a basis for comparison between the two groups.

## Limitations of the study

The follow-up data collection days for some of the pregnant women were not the same as their regular antenatal appointment days resulting in some of them not being available for follow-up data collection. Not being able to include the husbands of the pregnant women who participated in the interview also limited the study. The study could not match pregnant women with their partners. A qualitative study that will match pregnant women and their partners will provide more evidence on family involvement in BPCR.

## Conclusion

A group of pregnant women who were exposed to goal-oriented prenatal education had better knowledge and practice of BPCR. They also have higher number of institutional deliveries compared with their counterpart who were not. Therefore, goal-oriented prenatal education has proven to be effective at improving BPCR, as well as institutional delivery among pregnant

women. Implementation of goal-oriented prenatal education during antenatal care is hereby recommended.

## Acknowledgments

We appreciate the Consortium for Advanced Research and Training (CARTA) for supporting this research, and the University of Ibadan for creating an enabling environment to conduct this research. Special thanks to all the study participants, every member of staff of all health facilities involved in this study for their cooperation during data collection. We also appreciate all the research assistants.

## Author Contributions

**Conceptualization:** Margaret Omowaleola Akinwaare, Oyeninhun Abimbola Oluwatosin.

**Data curation:** Margaret Omowaleola Akinwaare.

**Formal analysis:** Margaret Omowaleola Akinwaare.

**Funding acquisition:** Margaret Omowaleola Akinwaare.

**Supervision:** Oyeninhun Abimbola Oluwatosin.

**Writing – original draft:** Margaret Omowaleola Akinwaare.

**Writing – review & editing:** Margaret Omowaleola Akinwaare,
Oyeninhun Abimbola Oluwatosin.

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
