## [Editor Report · Decision Letter 0]

9 Feb 2023

PONE-D-23-02382Effects of goal-oriented prenatal education on birth preparedness, complication readiness and institutional delivery among semi-urban pregnant women in Nigeria: a quasi-experimental studyPLOS ONE

Dear Dr. Akinwaare,

Thank you for submitting your manuscript to PLOS ONE. After careful consideration, we feel that it has merit but does not fully meet PLOS ONE’s publication criteria as it currently stands. Therefore, we invite you to submit a revised version of the manuscript that addresses the points raised during the review process.

We look forward to receiving your revised manuscript.

Kind regards,

Salisu Ishaku Mohammed, MD, MPH, MSc, PHD

Academic Editor

PLOS ONE

Journal Requirements:

3. Please ensure that you include a title page within your main document. We do appreciate that you have a title page document uploaded as a separate file, however, as per our author guidelines (http://journals.plos.org/plosone/s/submission-guidelines#loc-title-page) we do require this to be part of the manuscript file itself and not uploaded separately.

Additional Editor Comments:

•The title is very clear and comprehensive

•The abstract is precise. It describes what the issue is, what is being done about it, the how and end results. How the data was collected and analyzed was also briefly described. The conclusion is also clear and precise

•Introduction: The second sentence of the introduction (“the maternal mortality ratio of lifetime risk has been reported to be as high as 462 per 100,000 live births…..) should be revised to something like “the lifetime risk of death from pregnancy and delivery-related complications has been reported to be as high as 462 per 100,000 live births…….”).

In line 39 – 40, the authors said that “However, accessibility to skilled care during pregnancy and delivery has been associated with the prevention of maternal death”. Is it associated with prevention of maternal death or reduction of maternal death?

Line 40 – 42, “Thus, birth preparedness and complication readiness (BPCR) is a key strategy to receiving skilled care during pregnancy and delivery in developing countries like Nigeria”. Please, provide a reference here!

Line 64 -65, “Lack of adequate information on BPCR during prenatal education is associated with maternal death”. Please cite a reference here!

In general, the introduction section will make a great reading if further summarized and consolidated.

Methodology

Study design: Do we call a design with intervention and control groups to which women were randomly assigned a quasi-experimental?

Sample technique: In the abstract, the authors stated that “Four hundred pregnant women were randomly selected from two Local Government Areas of Ibadan Nigeria, and were randomized into an intervention group and control group”. In this section, the authors stated that (in line 94 – 95) “At the third stage, all antenatal clinic attendees who met the inclusion criteria at each visit were purposely selected to participate in the study”. Can the authors clarify how the selection was done? Random selection or purposeful selection?

Client recruitment and data collection: Are the research assistants who did the recruitment same as those that conducted the data collection? In order words, are the data collectors aware which intervention group the women belong?

Results

Table 1. is not very clear. The description of the variables in the table are not clear. For example, is the value for Age, the ‘mean age’ and the ‘standard deviation’? If this is the case, please revise as such. For other variables such as education, religion and level of education, are the number ‘n’ and percentages in bracket? Please indicate as appropriate. It is important to understand what those numbers indicate and how they are distributed between the intervention and control groups. And I believe that Table I and Table II can be merged into one to reduce the numbers of tables.

It is obvious that there are huge disparities in the distribution of some of the variables in the Tables between the control and the intervention groups (for example, the distribution of first trimester ANC registration, numbers of ANC visits before recruitment and so on). In these instances, could the authors perform X2 tests to indicate how significant the differences in these distributions? And before each Table is presented, could the authors briefly provide highlights on key findings in the Tables? I also believe it would be more reasonable to combine Table 3, 4 and 5 together.

Discussion

The discussion is great and detailed. However, the following adjustments should be considered:

•The firs paragraph should briefly describe what the study is all about and what is the overall success or otherwise associated with the study. No reference should be made to other studies in the first paragraph

•The authors should to only discuss the relevance of the findings on the study objectives, relationship to other studies without repeating the results in the discussion section. Only key findings (such BPCR and facility deliveries) should be discussed

•Try and identify one or two strengths of the study

•It is a good practice that the authors have mentioned a few limitations of the study, but only a technical deficiency should be considered as limitations. For example, the first limitation that the authors mentioned (prolongation of data collection period and the associated financial implications are not limitations in technical sense). Limitations are factors that weaken the study design that could bias the conclusion

Conclusion

The Conclusion can be summarized further to make it focused and concise. In addition, the main intervention being tested is the goal-oriented prenatal education and its effects on the predetermined outcomes. This should be described precisely in the conclusion.

---

## [Author Response · Author response to Decision Letter 0]

14 Feb 2023

Response to Reviewer’s Comments

The title page was added to the manuscript

The second sentence of the introduction which was previously stated as “the maternal mortality ratio of lifetime risk has been reported to be as high as 462 per 100,000 live birth was restated to read “the lifetime risk of death from pregnancy and delivery-related complications have been reported to be as high as 462 per 100,000 live births”

On line 68, the word “prevention” was replaced with “reduction”

Line 69, a new reference was added. The same reference was also added to the list of references on lines 432 – 434.

Line 93, a new reference was added. The same was added to the reference list on lines 453 – 455.

Study design: The women were not randomly allocated to the intervention and the control group. The randomization was done at the level of local government. The two selected local governments were randomized into intervention and control groups – line 117. Therefore, since the randomization of the women into intervention and control groups was not done, it is a quasi-experimental study.

Sampling technique: The sampling technique in the abstract section was revised as “Two Local Government Areas (LGAs) were randomly selected from the six semi-urban LGAs in Ibadan. These LGAs were randomized into an intervention and control group. Two Primary Healthcare Centres (PHCs) were randomly selected from each LGA, and 400 pregnant women who registered for antenatal care in the selected PHCs, and met the inclusion criteria were purposively selected to participate in the study” (lines 38 – 42). 

Client recruitment and data collection: The research assistants who did the recruitment also conducted the data collection. So, the data collectors were aware of which intervention group the women belong.

Results: Table 1 has been revised based, and it has also been merged with table 2. However, tables 3,4, and 5 could not be merged because each of them is targeting different specific objectives of the study. 

A brief introductory paragraph was added as the first paragraph in the discussion section.

The discussion section was revised to remove unnecessary repetition of results in the section.

The first two statements under the limitation for the study was deleted as suggested.

A section stating the strengths of the study was added (lines 396 – 402)

The conclusion section was revised to be more focused and concise.

---

## [Decision Letter · Decision Letter 1]

4 Jul 2023

PONE-D-23-02382R1Effect of goal-oriented prenatal education on birth preparedness, complication readiness and institutional delivery among semi-urban pregnant women in Nigeria: a quasi-experimental studyPLOS ONE

Dear Dr. Akinwaare,

Thank you for submitting your manuscript to PLOS ONE. After careful consideration, we feel that it has merit but does not fully meet PLOS ONE’s publication criteria as it currently stands. Therefore, we invite you to submit a revised version of the manuscript that addresses the points raised during the review process.The title is okay. The following points should be addressed as necessary for acceptance of the manuscript: The major problem statement you are addressing needs to be clearly highlighted, the inclusion and exclusion criteria and potential biases need to be addressed as has been stated by one of the reviewers,  typographical errors need to be corrected and ensure that the referencing style for all your references are in in line with the journal's requirements. Please address the concerns raised by the reviewer. 

We look forward to receiving your revised manuscript.

Kind regards,

Adaoha Pearl Pearl Agu, MBBS, MSc, FMCPH

Academic Editor

PLOS ONE

Journal Requirements:

Reviewers' comments:

Reviewer's Responses to Questions

**Comments to the Author**

1. If the authors have adequately addressed your comments raised in a previous round of review and you feel that this manuscript is now acceptable for publication, you may indicate that here to bypass the “Comments to the Author” section, enter your conflict of interest statement in the “Confidential to Editor” section, and submit your "Accept" recommendation.

Reviewer #1: All comments have been addressed

Reviewer #2: All comments have been addressed

2. Is the manuscript technically sound, and do the data support the conclusions?

Reviewer #1: Yes

Reviewer #2: Partly

3. Has the statistical analysis been performed appropriately and rigorously? 

Reviewer #1: Yes

Reviewer #2: Yes

4. Have the authors made all data underlying the findings in their manuscript fully available?

Reviewer #1: Yes

Reviewer #2: Yes

5. Is the manuscript presented in an intelligible fashion and written in standard English?

Reviewer #1: Yes

Reviewer #2: Yes

6. Review Comments to the Author

Reviewer #1: The paper has really approved from the previous version. The authors have adequately addressed the comments and the document now reads better. No further comment

Reviewer #2: Thank you for the opportunity to review this manuscript. This is well done and your research results are very encouraging if this type of approaches are applied in the country it is very likely to have an impact on reducing MMR.

Professional English editing is needed.

The authors should revise the introduction to clearly highlight the problem statement. Together, I would suggest the authors during their write up to consider using recent BPCR references from countries with similar maternal mortality rates.

The entire method section requires revision to improve readability by ensuring there is clear flow of information. Please include a discussion on inclusion and exclusion criteria. How many participants were not included due to not having a cell phone? Was this due to socio-economic status and was this an exclusion that could inherently bias the sample to wealthier and possibly more educated patients? Was the sample stratified by the number of previous pregnancies? This too could potentially bias the analysis. Line 127-133, the authors state: "adopted a validated, semi-structured/structured questionnaire from 'Monitoring BPCR tools and indicators for maternal and new-born health' developed by Johns Hopkins Program for International Education in Gynaecology and Obstetrics” They should indicate what adaptations or changes were made to the BP/CR tool in the local context.

The study does not give clear implications in the conclusion or recommendations. If the study provides similar findings as other studies, where is the problem? What is the explanation for this? What should be done about this situation? What should be the next logical step to address this issue?

The manuscript needs revision for some typographical errors as well need to revise the title

The authors should make sure you use reference style for this journal

7. PLOS authors have the option to publish the peer review history of their article (what does this mean?). If published, this will include your full peer review and any attached files.

Reviewer #1: **Yes: **James Orwa

Reviewer #2: **Yes: **Richard Kalisa

---

## [Author Response · Author response to Decision Letter 1]

15 Jul 2023

Response to reviewer’s comments

Introduction section was reviewed to highlight problem statements

The inclusion criteria were discussed at the methods section

Using of a cell phone was not a criterium for inclusion

The sample was not stratified by the number of previous pregnancies

The validated questionnaire developed by Johns Hopkins Program for International Education in Gynaecology and Obstetrics was “adopted” and not “adapted” with little or no changes. Except for change of terminologies to simple terms that can be easily understood by the study participants.

Comparison with similar previous studies were highlighted in the paper, with relevant implications.

The manuscript was revised for some typographical errors

Reference style for the journal was ensured.

---

## [Editor Report · Decision Letter 2]

19 Jul 2023

Effect of goal-oriented prenatal education on birth preparedness, complication readiness and institutional delivery among semi-urban pregnant women in Nigeria: a quasi-experimental study

PONE-D-23-02382R2

Dear Dr.Akinwaare,

We’re pleased to inform you that your manuscript has been judged scientifically suitable for publication and will be formally accepted for publication once it meets all outstanding technical requirements.

Kind regards,

Adaoha Pearl Agu, MBBS, MSc, FMCPH

Academic Editor

PLOS ONE
---

## [Editor Report · Acceptance letter]

21 Jul 2023

PONE-D-23-02382R2 

 Effect of goal-oriented prenatal education on birth preparedness, complication readiness and institutional delivery among semi-urban pregnant women in Nigeria: a quasi-experimental study 

Dear Dr. Akinwaare:

I'm pleased to inform you that your manuscript has been deemed suitable for publication in PLOS ONE. Congratulations! Your manuscript is now with our production department. 

Kind regards, 

on behalf of

Dr. Adaoha Pearl Pearl Agu 

Academic Editor

PLOS ONE